

# Evaluation of a toxoid fusion protein vaccine produced in plants to protect poultry against necrotic enteritis

Joseph G.L. Hunter[1,2,*], Shyra Wilde[1,*], Amanda M. Tafoya[1], Jamie Horsman[1], Miranda Yousif[1], Andrew G. Diamos[1,2], Kenneth L. Roland[1] and Hugh S. Mason[1,2]

[1] Center for Immunotherapy, Vaccines and Virotherapy, The Biodesign Institute, Arizona State University, Tempe, AZ, USA
[2] School of Life Sciences, Arizona State University, Tempe, AZ, USA
* These authors contributed equally to this work.

## ABSTRACT

**Background:** Necrotic enteritis (NE) is caused by type A strains of the bacterium *Clostridium perfringens*. Total global economic losses to the poultry industry due to NE is estimated to be over two billion dollars annually. Traditionally, NE has been effectively controlled by inclusion of antibiotics in the diet of poultry. However, recent concerns regarding the impact of this practice on increasing antibiotic resistance in human pathogens have led us to consider alternative approaches, such as vaccination, for controlling this disease. NE strains of *C. perfringens* produce two major toxins, a-toxin and NetB. Immune responses against either toxin can provide partial protection against NE.
**Methods:** We have developed a fusion protein combining a non-toxic carboxyl-terminal domain of a-toxin (PlcC) and an attenuated, mutant form of NetB (NetB-W262A) for use as a vaccine antigen to immunize poultry against NE. We utilized a DNA sequence that was codon-optimized for *Nicotiana benthamiana* to enable high levels of expression. The 6-His tagged PlcC-NetB fusion protein was synthesized in *N. benthamiana* using a geminiviral replicon transient expression system, purified by metal affinity chromatography, and used to immunize broiler birds.
**Results:** Immunized birds produced a strong serum IgY response against both the plant produced PlcC-NetB protein and against bacterially produced His-PlcC and His-NetB. Immunized birds were significantly protected against a subsequent in-feed challenge with virulent *C. perfringens* when treated with the fusion protein. These results indicate that a plant-produced PlcC-NetB toxoid is a promising vaccine candidate for controlling NE in poultry.

## INTRODUCTION

*Clostridium perfringens*, a Gram-positive anaerobic spore-forming bacterium, is the causative agent of necrotic enteritis (NE) in poultry and has become an economically

Corresponding author
Hugh S. Mason,
hugh.mason@asu.edu

significant issue for the broiler industry (*Timbermont et al., 2011*). The pathogenesis of NE is complex and results from the concerted actions of a number of *Clostridial* genes combined with reduced gut health generated by environmental factors, feed, mycotoxins or the presence of other pathogens (*Prescott et al., 2016*; *Shivaramaiah et al., 2011*). NE typically occurs in broiler chicks at two-six weeks of age, but may occur in older birds as well (*Cooper & Songer, 2010*; *Cooper, Songer & Uzal, 2013*). The onset of NE at around two weeks is believed to be due to waning levels of anti-Clostridium maternal antibodies (*Cooper, Songer & Uzal, 2013*). The acute form of the disease leads to increased mortality in broiler flocks, which can account for high losses of up to 1% per day, reaching mortality rates up to 10–40% (*McDevitt et al., 2006*). In the subclinical form, damage to the intestinal mucosa caused by *C. perfringens* leads to poor productivity (reduced growth, reduced feed efficiency) without mortality. The estimated cost of subclinical NE is around $0.05 per chicken with a total global loss of nearly $2 billion per year (*McReynolds et al., 2004*; *Van Der Sluis, 2000*).

*Clostridium perfringens* is ubiquitous; readily found in soil, dust, feces, feed, used poultry litter and is normally present in the intestines of healthy chickens. The enterotoxaemia that results in clinical disease most often occurs either following an alteration in the intestinal microflora or from a condition that results in damage to the intestinal mucosa (e.g., coccidiosis). High dietary levels of animal by-products such as fishmeal, or certain grains with high levels of indigestible fiber including wheat, barley, oats, and rye predispose birds to NE. Dietary fat source can also influence the *C. perfringens* population (*Knarreborg et al., 2002*). Current thinking is that predisposing factors such as a high protein diet (e.g., fishmeal) and/or *Eimeria* infection result in alterations of the chicken gut microbiota that allow incoming or resident *C. perfringens* strains to become established (*Stanley et al., 2014*).

Historically, *C. perfringens* outbreaks in the broiler industry were avoided using growth-promoting antimicrobials in the diet. However, concerns regarding antibiotic resistance led to restrictions on the use of antibiotics (*Butaye, Devriese & Haesebrouck, 2003*). When antibiotics were banned in European and Scandinavian countries, it led to increases in *C. perfringens* infections, including NE (*Van Immerseel et al., 2004*). Thus, other methods for controlling *C. perfringens* infections are needed.

Toxins have traditionally been targeted as antigens of interest for controlling clostridial infections. The *C. perfringens* α-toxin, encoded by the *plc* gene, is the major virulence determinant for gas gangrene (*Titball, Naylor & Basak, 1999*). Antibodies to *C. perfringens* α-toxin prevent gas gangrene in mice (*Williamson & Titball, 1993*). The protein is divided into two domains; the amino-terminal domain and carboxy-terminal domain (*Naylor et al., 1998*). The amino-terminal domain encodes the catalytic site responsible for phospholipase activity (*Naylor et al., 1998*; *Guillouard, Garnier & Cole, 1996*; *Titball et al., 1991*), while the carboxy-terminal domain is involved in interactions with phospholipids, targeting the enzyme to host cell membranes (*Guillouard et al., 1997*). The α-toxin carboxy-terminal fragment (aa 247–370) is non-toxic (*Titball, Fearn & Williamson, 1993*) and immunization with this fragment confers protection against α-toxin and *C. perfringens* in a gas gangrene mouse model (*Williamson & Titball, 1993*).

In chickens, immune responses against the C-terminal domain (PlcC) can provide partial protection against intestinal lesions following challenge with *C. perfringens* (*Williamson & Titball, 1993*; *Jiang et al., 2015*; *Kulkarni et al., 2010*; *Zekarias, Mo & Curtiss, 2008*).

Based on its association with gas gangrene, the chromosomally-encoded α-toxin was considered to be the major toxin associated with NE (*Titball, Naylor & Basak, 1999*; *Williamson & Titball, 1993*; *Al-Sheikhly & Truscott, 1977*; *Fukata et al., 1988*). The C-terminal domain of the α-toxin (PlcC) has been studied extensively as a vaccine against *C. perfringens* infection, delivered as a purified protein (*Williamson & Titball, 1993*; *Stevens et al., 2004*; *Nagahama et al., 2013*) or by live attenuated bacteria (*Zekarias, Mo & Curtiss, 2008*; *Hoang et al., 2008*). Currently the only commercially available vaccine for NE is composed of an α-toxoid derived from a *C. perfringens* type A strain (*Crouch et al., 2010*).

More recent studies have identified a β-like toxin, designated NetB toxin, linked to NE. It was originally identified in a *C. perfringens* type A strain isolated from an Australian outbreak of NE (*Keyburn et al., 2008*). *Keyburn et al. (2008)* showed that *C. perfringens* supernatants contain a factor that is toxic for chicken leghorn male hepatoma cells. The factor was present in supernatant from a *plc* knockout mutant, but not a *netB* knockout mutant of *C. perfringens*, implicating NetB as the toxin. Further, the *C. perfringens netB* mutant was attenuated for virulence in chickens, whereas a *plc* mutant of the same parent strain retained full virulence (*Keyburn et al., 2006*). Screens for *netB* in *C. perfringens* field isolates indicate that the presence of the *netB* gene is highly correlated with NE causing strains (*Keyburn et al., 2008*, *2010*; *Tolooe et al., 2011*). Thus, NetB has emerged as a critical virulence factor. NetB functions by binding to cholesterol in membranes, forming heptameric pores (*Savva et al., 2013*; *Yan et al., 2013*). There are a number of single amino acid substitutions in the rim loop region of the protein, believed to interact with the host membrane, that significantly reduce its toxicity. These include Y191A, R200A, W257A and W262A (*Savva et al., 2013*) and S254L, R230Q and W287R (*Yan et al., 2013*).

NetB is also a protective antigen, providing partial protection against *C. perfringens* challenge, particularly in combination with other immunogenic components (*Da Costa et al., 2013*; *Keyburn et al., 2013a*, *2013b*). In addition, immunization with non-toxic rim mutants, W262A (*Da Costa et al., 2013*) and S254L (*Keyburn et al., 2013a*), have been shown to generate protective immune responses in chickens. A study that examined serum antibody levels against *C. perfringens* α-toxin and NetB toxin in commercial birds from field outbreaks of NE showed that the levels of serum antibodies against both α-toxin and NetB toxin were significantly higher in apparently healthy chickens compared to birds with clinical signs of NE, suggesting that these antitoxin antibodies may play a role in protection (*Lee et al., 2012*). These results indicate a correlation between the presence of antitoxin antibodies in the serum and protective immunity against NE.

In a previous study, we used a live attenuated *Salmonella* vaccine to deliver PlcC, NetB, or both to chickens. We found that birds immunized against both toxins developed fewer severe lesions after *C. perfringens* challenge than birds immunized with either antigen alone (*Jiang et al., 2015*). Based on this finding, we have been developing approaches to combine the immunogenic components of each toxoid into a single protein, in order to
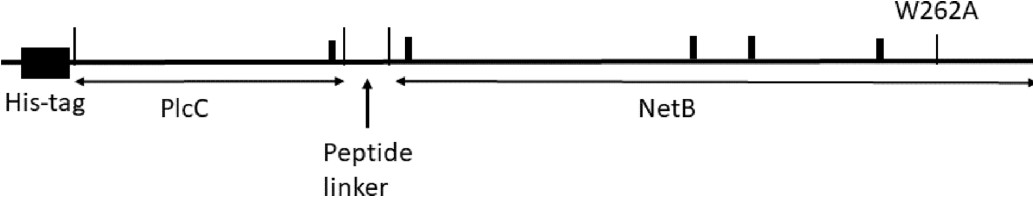

**Figure 1 Diagram of the 6His-plcC-netB fusion protein.**

develop a more convenient and protective vaccine for NE. In this study, we describe a novel *plcC-netB* fusion gene modified to encode the rim mutation W262A and codon-optimized for production in *Nicotiana benthamiana*. The gene was introduced into *N. benthamiana* via *Agrobacterium tumefaciens*-mediated transformation of a geminiviral replicon transient expression system. The resulting fusion protein was purified and used to inject broiler chicks. Results indicate that the protein was highly immunogenic and elicited partial protection against challenge with a virulent strain of *C. perfringens*.

## MATERIALS AND METHODS

### Expression vector construction

A PlcC-NetB fusion protein gene was designed using codons preferred for *N. benthamiana* (*Geyer et al., 2010*). Amino acid sequences were based on GenBank accessions AAP15462.1 for PlcC and ABW71134.1 for NetB. For targeting of nascent protein to the endoplasmic reticulum (ER), the barley alpha amylase signal peptide, followed by a 6His tag, was added to the N-terminus of PlcC. A linker peptide GGSGGSGGPSGGSGG was inserted between the PlcC and NetB sequences (Fig. 1). The fusion protein nucleotide sequences are in GenBank accession numbers MH883808 and MH883809. The gene was synthesized commercially with flanking XbaI and SacI sites. The 1,402 bp XbaI-SacI fragment was ligated with geminiviral vector pBYR2eK2M (*Diamos, Rosenthal & Mason, 2016*) to make pBYR2eK2M-6HplcCnetB. Removal of the 417 bp BamHI fragment containing the PlcC and linker sequences yielded the ER-targeted 6His-NetB construct pBYR2eK2M-6HnetB. To make the ER-targeted 6His-PlcC vector, we PCR end-tailored the PlcC sequence to introduce a stop codon and SacI site at its 3′ end, using the primer PlcC-Sac-R (Table S1), inserting the resulting product to yield pBYR2eK2M-6HplcC. To produce the cytosol-targeted proteins, the signal peptide was removed by introducing an XbaI site and start codon 5′ of the 6His sequence by PCR with primer 6H-Xba-F and using the ER-targeting constructs above as templates. The resulting XbaI-SacI fragments were ligated with plasmid pBYR2e3K2Mc-GFP (*Diamos & Mason, 2018*; *Huang & Mason, 2004*) digested XbaI-SacI, thus yielding the vectors pBYR2e3K2Mc-c6HplcCnetB, pBYR2e3K2Mc-c6HplcC, and pBYR2e3K2Mc-c6HnetB.

### Agroinfiltration of plants for protein production

Expression vectors were inserted into *A. tumefaciens* EHA105 via electroporation. PCR and restriction digestion of the purified plasmids were used to confirm the resulting strains. *Agrobacterium* strains were grown overnight at 30 °C in YENB (7.5 g Yeast Extract

and eight g Nutrient Broth for one L) + 50 mg/L kanamycin and 2.5 mg/L rifampicin. Bacteria were pelleted for 10 min at 5,000 g, then resuspended in infiltration buffer (10 mM 2-(N-morpholino) ethanesulfonic acid (MES), pH 5.5 and 10 mM $MgSO_4$) with a final OD600 = 0.2. Suspensions were infiltrated into *N. benthamiana* leaves at 5- to 6-weeks of age using a syringe without needle via a small puncture (*Huang & Mason, 2004*). Infiltrated leaves were harvested at 5 days post infiltration (DPI).

## Protein extraction, acid precipitation clarification, and filtration

Total protein was extracted from agroinfiltrated leaf samples using a 1:3 (*w:v*) ratio of extraction buffer (50 mM sodium phosphate, 200 mM sodium chloride, pH 8.1) at 4 °C. Leaf samples were homogenized for 5–10 min until an even consistency was obtained using a two speed Conair™ Waring™ Laboratory Blender (Thermo Fisher Scientific, Waltham, MA, USA). Homogenized leaf tissue was stirred at high speed for 10 min at 4 °C to improve protein solubility.

Acid precipitation was performed on crude extracts after this 10-min stirring period. Note that no filtration, sedimentation, or centrifugation was performed before acid precipitation. One M phosphoric acid was added to the crude extract as it spun until a pH of 4.4–4.6 was reached, as measured by a VWR Sb70 Symphony Ph Meter (VWR, Radnor, PA, USA), Extract was stirred for 10 min at 4 °C. Two M Tris base was then added to the extract until pH reached 7.5–7.7, and the mixture was stirred for another ten min at 4 °C before being filtered through Miracloth and centrifuged at 17,000 g, 4 °C, for 30 min. The extract was then subjected to a second filtration through Miracloth resulting in a clarified extract, which was passed through a 0.45-micron filter for sterilization before metal affinity chromatography.

Three mL of Talon metal affinity resin (Clontech/Takara Bio, Mountain View, CA, USA) was spun down at 10,000 g, 4 °C, for 5 min and the supernatant removed from the bed. The resulting 1.5 mL resin bed was washed using 15 mL of extraction buffer (referred to below as wash buffer) and once again spun down at 10,000 g, 4 °C, for five min. Supernatant was discarded and the resin resuspended in 15 mL wash buffer before being added to a 20 mL column. A final wash of 20 mL was performed before the filtered extract was passed through the column. The resin was washed with 60 mL of wash buffer, and elution of bound proteins achieved with PBS + 150 mM imidazole, pH 7.4, collected in five 1 mL samples. Elution samples were dialyzed overnight in one L of sterile PBS using either a three mL or 10 mL 3,500 MWCO Slide-A-Lyzer G2 dialysis cassette (Thermo Fisher Scientific, Waltham, MA, USA), following manufacturer's instructions. Elution concentrations were determined both before and after dialysis by an $A_{280}$ absorption using PBS and PBS + 150 mM imidazole, respectively, as blank solutions. Protein concentrations were confirmed by Coomassie staining of SDS-PAGE (below) analyzed with ImageJ (*Schneider, Rasband & Eliceiri, 2012*).

## SDS-PAGE and Western blot

Crude extracts, clarified extracts, and eluted samples were mixed with SDS sample buffer (50 mM Tris-HCl, pH 6.8, 2% SDS, 10% glycerol, 0.02% bromophenol blue) under

reducing conditions (0.5 M DTT) and samples boiled for 10 min and separated on 4–15% polyacrylamide gels (Bio-Rad, Hercules, CA, USA). Gels were stained with Coomassie stain (Bio-Rad, Hercules, CA, USA) or electroblot transferred to a PVDF membrane. PVDF membranes were blocked with 5% PBSTM (PBS with 0.05% Tween-20 and 5% nonfat dry milk) overnight at 4 °C. Membranes were then washed with PBST before incubation with a 1:10,000 dilution of rabbit anti-PlcC or rabbit anti-NetB antibodies (*Kulkarni et al., 2010*) in 1% PBSTM for 1 h at 37 °C. After washing a second time the membranes were incubated with a 1:10,000 dilution of goat anti-rabbit horseradish peroxidase conjugate in 1% PBSTM for 1 h at 37 °C. Bound antibody was detected with ECL reagent (Amersham Biosciences, San Francisco, CA, USA). Stripping of blots for reprobing was performed using a modified Abcam protocol. PVDF membrane was washed for 10 min in mild stripping buffer (for one L: 15 g glycine, one g SDS, 10 mL Tween20, pH 2.2) per Abcam's recipe. PVDF membrane was then rinsed twice in 20 mL of PBS for 10 min followed by two 5-min rinse steps in 20 mL PBST. 5% PBSTM was used to block the membrane before reprobing.

## Production of recombinant proteins in *E. coli* for immune assays

His-tagged recombinant PlcC (rPlcC) and NetB (rNetB) were cloned by PCR using primer sets Plc_pqe_f1/Plc_pqe_r2 and netB_pqe_f3/netB_pqe_r4, respectively (Table S1), using pYA5130 as the template. The resulting PCR fragments were digested with BamHI and SalI and ligated to BamHI/SalI-digested plasmid pQE30. Plasmids carrying the desired inserts were moved into *E. coli* strain M15(pREP4) (Qiagen, Hilden, Germany). Protein synthesis was induced by the addition of IPTG according to the manufacturer's instructions. rPlcC and rNetB were purified by passage over a TALON affinity column (Clontech Laboratories, Inc, Mountain View, CA, USA) following the manufacturer's recommendations. The GST-NetB protein shown in Figs. 2 and 3 was purified as previously described (*Jiang et al., 2015*).

## Chicken experiments

All animal experiments were conducted in compliance with the Arizona State University Institutional Animal Care and Use Committee (Protocol 16-1480R) and the Animal Welfare Act, in accordance with the National Institutes of Health guide for the care and use of laboratory animals (NIH Publication No. 8023, revised 1978).

Groups of one-day-old Cornish × Rock broiler chickens were purchased from the Murray McMurray Hatchery (Webster City, IA, USA) and randomly sorted and placed in pens with pine shavings on the floor. Food and water was supplied *ad libitum*. For Experiments 1 and 2, groups of birds were immunized subcutaneously with 50 μg of fusion protein (Exp. 1, seven vaccinated and 11 control birds) or 100 μg of fusion protein (Exp. 2, 13 vaccinated and 10 control birds) with 50 μg of Quil-A adjuvant. Birds were given three immunizations at weekly intervals beginning at 8 days of age, with the final immunization at 22 days of age. Mock vaccinated controls were injected with adjuvant only. At 27 days of age, blood was taken for serum analysis. Three days later, when the birds were 30 days of age, birds were given an in-feed challenge with virulent *C. perfringens*

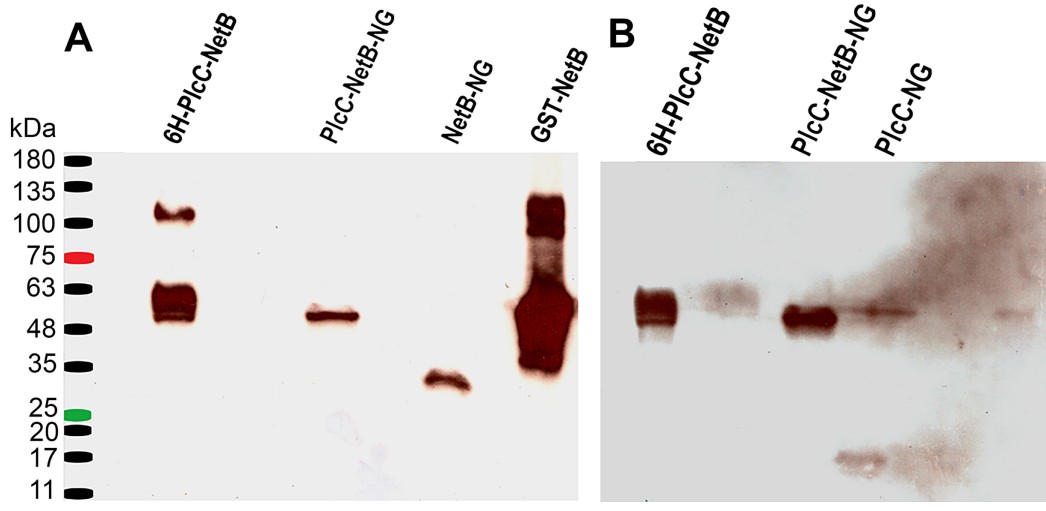

**Figure 2 Western blot comparing different fusion constructs.** (A) Blot probed with rabbit anti-NetB. (B) Blot used in (A) after stripping, probed with rabbit anti-PlcC. 6H-PlcC-NetB, ER targeted fusion protein expressed in *N. benthamiana*leaf; PlcC-NetB-NG, cytosolically targeted fusion protein expressed in *N. benthamiana*leaf; NetB-NG, cytosolically targeted NetB expressed in *N. benthamiana*leaf; GST-NetB, bacterially expressed standard protein; PlcC-NG, cytosolically targeted PlcC expressed in *N. benthamiana*leaf.

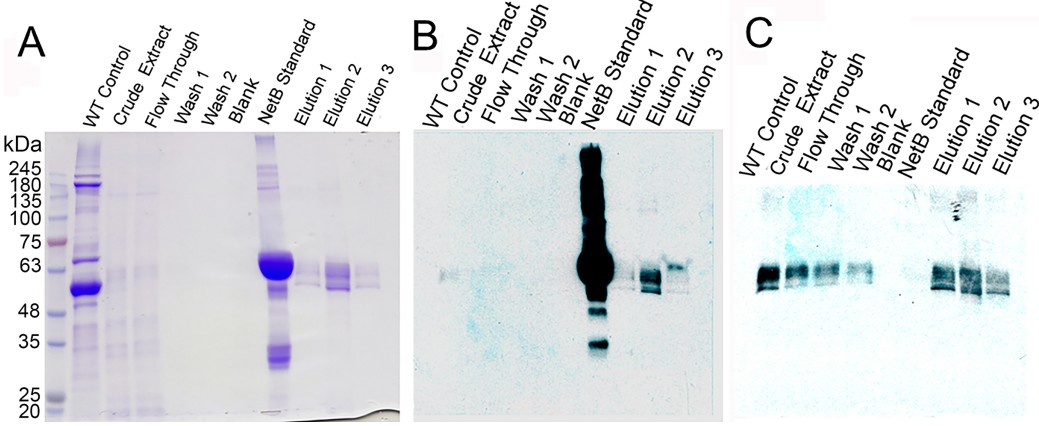

**Figure 3 Purification of glycosylated PlcC-NetB fusion.** (A) Coomassie-stained gel showing all purification steps. (B) Western blot of all purification steps probed with rabbit anti-NetB. (C) Western blot of all purification steps probed with rabbit anti-PlcC. Note: NetB standard is a GST-NetB fusion with a mass of 59.9 kDa (*Jiang et al., 2015*).

for 5 days. The birds were scored for intestinal lesions the next day, at 35 days of age. For Experiment 3, birds were injected with 50 µg of PlcC-NetB at 7 days of age (22 vaccinated and 19 controls) and again at 12 days of age. Blood and tissues were taken from five birds/group for immunological analysis 10 days after the boost. Birds were challenged 2 weeks after the final boost, when the birds were 26 days of age.

## Enzyme-linked immunosorbent assay (ELISA)

ELISAs were performed in triplicate as described *Williamson & Titball (1993)* to determine the IgY responses against his-tagged PlcC, his-tagged Fba, his-tagged NetB and

S. Typhimurium lipopolysaccharide in chicken sera and IgA, IgY and IgM responses in intestinal washes. Biotinylated anti-chicken IgA (Alpha Diagnostic Intl. Inc., San Antonio, TX, USA), IgY (Southern Biotechnology, Birmingham, AL, USA) or IgM (Bioss, Woburn, MA, USA) antibodies diluted 1:10,000 were used to detect the various antibody isotypes.

## Cellular proliferation assay

A proliferation assay was performed to evaluate cell-mediated immunity. Lymphocytes in the blood were harvested using the gentle swirl technique (*Timbermont et al., 2011*) and plated in quadruplicate, in a 96-well plate at $10^5$ cells/well in RPMI-1640 without phenol red. Spleens were placed through a 70 μm cell strainer to obtain single cell suspensions. Red blood cells were lysed with Red Blood Cell Lysis solution (eBioscience, San Diego, CA, USA). Splenocytes were then washed, suspended in RPMI and plated at $10^6$ cells/well. Each set of cells was incubated at 37 °C, 5% $CO_2$ for 72 h with or without four μg/mL of either His-Fba, His-NetB, His-PlcC, or one μg/mL PMA. Cell proliferation was measured using the Vybrant®MTT Cell Proliferation Assay Kit (Molecular Probes, Eugene, OR, USA). Mean absorbance value of antigen stimulated wells divided by mean absorbance of non-stimulated control wells was used to calculate stimulation index.

## Challenge with *C. perfringens*

Chickens were fed an antibiotic-free starter feed containing 21% protein for 20 days, at which time the feed was switched to a high protein (28% protein), wheat-based feed containing 36% fish meal and zinc at 400 ppm (customized by Reedy Fork Farm, Elon, NC, USA) to predispose the birds to NE (*Coursodon et al., 2012*; *Shojadoost, Vince & Prescott, 2012*). Birds were challenged with virulent *C. perfringens* strain CP4 (*Thompson et al., 2006*) in feed for 5 days as described (*Shojadoost, Vince & Prescott, 2012*) with some modifications. Feed was withdrawn on the day prior to challenge for 15 h. The following day, chickens were inoculated with 1.0 mL of an overnight culture of *C. perfringens* CP4 grown in cooked meat medium (CMM; Difco, Detroit, MI, USA) by oral gavage. Immediately after gavage, infected feed was provided thereafter for five consecutive days. To prepare infected feed, *C. perfringens* was grown in CMM for 24 h at 37 °C, which then was inoculated into fluid thioglycollate medium (Difco) at a ratio of 0.3% (v/v) and incubated at 37 °C for 15 h (morning challenge) or 23 h (evening challenge). The *C. perfringens* culture was mixed with feed at a ratio of 1:1 (v/w). Infected feed was prepared freshly twice daily. All birds were euthanized and necropsied the day following the final challenge.

## Lesion scoring

Protection against *C. perfringens* challenge was assessed on the basis of gross intestinal lesion scores at necropsy. On day 33, chickens were euthanized with $CO_2$ and their small intestines (defined here as the section between the gizzard and Meckel's diverticulum) were examined for visible gross lesions. Intestinal lesions were scored as follows: 0 = no gross lesions; 1 = thin or friable wall or diffuse superficial but removable fibrin; 2 = focal necrosis or ulceration, or non-removable fibrin deposit, 1–5 foci; 3 = focal necrosis or ulceration, or non-removable fibrin deposit, 6–15 foci; 4 = focal necrosis or ulceration,

or non-removable fibrin deposit, 16 or more foci; 5 = patches of necrosis two to three cm long; 6 = diffuse necrosis typical of field cases (*Shojadoost, Vince & Prescott, 2012*). Birds that died during challenge were assigned a score of 6.

### Statistical analysis

All statistics were carried out using GraphPad Prism 6.0 (Graph-Pad Software, San Diego, CA, USA). Antibody titers and lesion scores were analyzed using one-way ANOVA followed by Tukey's posttest. The values were expressed as means ± SEM, and differences were considered significant at $P < 0.05$.

## RESULTS

### Construction of a PlcC-NetB fusion protein

We designed a fusion protein comprising an N-terminal 6His tag, PlcC, linker GGSGGSGGPSGGSGG, and NetB (Fig. 1). The long, flexible linker was used to facilitate correct folding of the PlcC and NetB domains, and to permit unencumbered access of B-cell receptors to the important immunogenic regions. PlcC contains the C-terminal receptor-binding domain of the alpha toxin (*Guillouard et al., 1997*). The NetB sequence contains a mutation W262A in the rim domain (*Savva et al., 2013*), which substantially attentuates the toxicity of the protein in cell culture assays. We noted that PlcC and NetB contain a total of five eukaryotic Asn-linked glycosylation sites (N-X-S/T), which could result in the addition of glycans to the Asn residue when the protein is targeted to the ER in plant cells. We constructed expression vectors to target proteins either to the ER (pBYR2eK2M-6HplcCnetB, -6HplcC and -6HnetB) or to the cytosol (pBYR2e3K2Mc-c6HplcCnetB -c6HplcC, and -c6HnetB). We expected the cytosol targeted proteins to be unglycosylated (designated PlcC-netB-NG), while the ER-targeted proteins may be glycosylated at any combination of or all of the five consensus N-X-S/T sites.

### Production of PlcC-NetB and PlcC-NetB-NG fusion proteins in plants

Using the methods outlined herein two different fusion proteins were produced. One, targeted to the ER, was glycosylated (PlcC-NetB). The other, targeted to the cytoplasm, was non-glycosylated (PlcC-NetB-NG). Both fusion proteins were probed with rabbit anti-PlcC and rabbit anti-NetB antibodies which yielded the same results on a western blot (Fig. 2). PlcC-NetB also has incrementally increased band sizes above the initial band which aligns with the singular PlcC-NetB-NG band at approximately 49 kDa, the expected relative size of the fusion proteins (Fig. 2). Both NetB and PlcC expressed individually were also probed and appear at the expected sizes of 34 kDa and 15 kDa, respectively. While ER and cytoplasmic targeted NetB and PlcC were produced, only representative results are shown in Fig. 2. The yield of all PlcC-NetB constructs was estimated to be ~2 mg/g leaf mass (~20% of total soluble protein).

The process of purification for the PlcC-NetB protein is outlined in Fig. 3. A Coomassie-stained gel (Fig. 3A) shows a wild-type, uninfiltrated control followed by crude extract which had already been acid precipitated. Note that the crude extract has highly reduced plant protein contaminants, due to the acid precipitation that caused contaminating

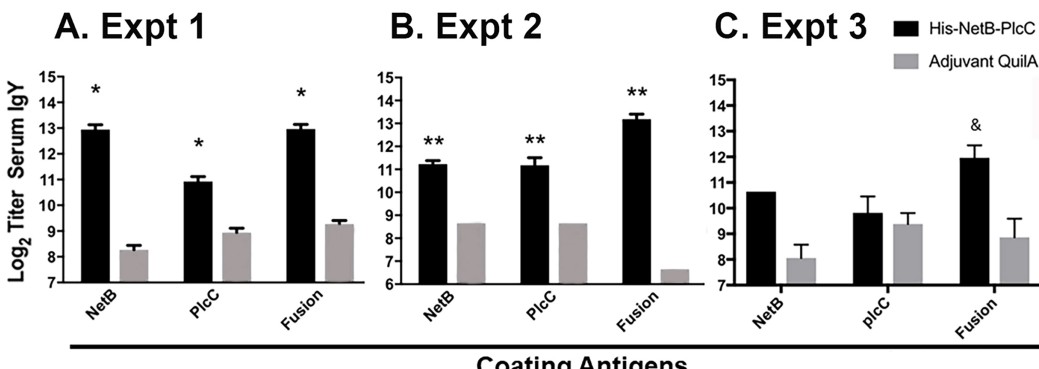

**Figure 4 IgY serum titers from immunized and non-immunized birds, assayed against NetB, PlcC, or NetB-PlcC fusion.** All *P*-values are compared to adjuvant only controls. Fusion refers to his-tagged, glycosylated PlcC-NetB. (A) Experiment 1, *$P \leq 0.009$ (B) Experiment 2, **$P \leq 0.0025$. (C) Experiment 3, &$P < 0.0002$.

proteins to become insoluble while the fusion proteins remain soluble. The flow-through fraction contained bands of a similar profile to the crude extract on the Coomassie, while both washes showed no detectable protein. Three elution fractions contained bands at approximately 49 kDa. Anti-NetB antibodies (Fig. 3B) reacted primarily with the standard and elution 2, while elutions 1 and 3 had visible but less prominent bands. When probed with anti-PlcC antibodies (Fig. 3C), similar bands were present in every lane except the blank and NetB standard lanes. Elution fraction bands were as strong as those seen in the crude extract lane. Flow-through and wash fractions had lighter but detectable bands when probed with anti-PlcC. Bands that reacted with anti-PlcC antibodies were also seen at approximately 49 kDa.

## Serum antibody responses to fusion protein antigens

We evaluated the serum antibody responses in birds immunized with PlcC-NetB and in control birds against the glycosylated fusion protein and against non-glycosylated recombinant rPlcC and rNetB produced in bacteria. In Experiments 1 and 2, the anti-PlcC-NetB titers were significantly higher than controls (Fig. 4). The titers against rPlcC and rNetB were also significantly elevated over controls. In Experiment 3, where birds were immunized twice, only the titers against the fusion protein were significantly higher than controls, although the serum titers against rNetB were elevated.

## Cellular responses against PlcC-NetB in Experiment 3

Immunization with the fusion protein led to variable cellular responses. The rPlcC-NetB fusion protein elicited weak, but significant, cellular responses against rNetB and rPlcC-NetB in splenocytes (Fig. 5). No antigen-specific stimulation was detected in lymphocytes (data not shown).

## Protection against *C. perfringens* challenge

Protection against challenge was assessed by scoring intestinal lesions using a six-point scoring system (*Shojadoost, Vince & Prescott, 2012*). Immunization with the

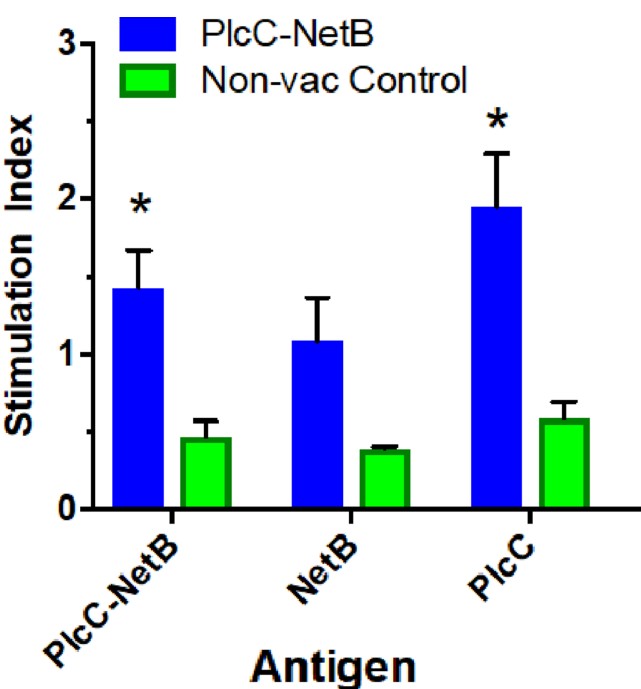

**Figure 5 Splenocyte responses.** Splenocyte responses to PlcC-NetB, PlcC and NetB after immunization with PlcC-NetB in Fig. 4, Experiment 3. *$P$ < 0.02. 

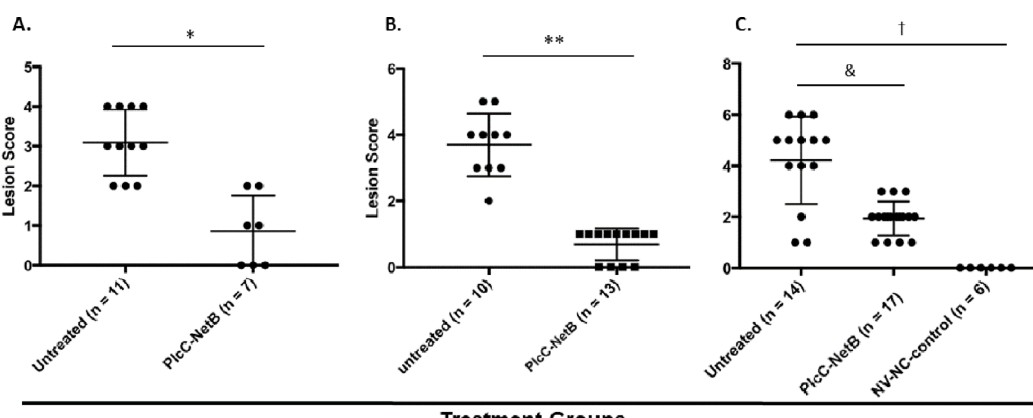

**Figure 6 Intestinal lesions scores.** Intestinal lesions scores in (A) Experiment 1, (B) Experiment 2, and (C) Experiment 3. Each dot represents the score of an individual bird. Differences between immunized groups and controls are indicated. *$P$ < 0.001; **$P$ = 0.004; &$P$ = 0.0006; †$P$ < 0.0001.

rPlcC-NetB fusion protein resulted in a significant reduction in lesion scores in all three experiments (Fig. 6). Note that in Experiment 3, birds were challenged a week earlier than in Experiments 1 and 2. The younger age of the birds may have contributed to the higher lesion scores for some birds. Two of the non-immunized birds died during challenge. This was the only experiment in which any birds died following challenge.

## DISCUSSION

There is a growing need for methods to control NE that do not rely on antibiotics. Vaccination is a proven approach for controlling many poultry diseases and immunization with toxoid antigens is known to provide some protection against NE (*Keyburn et al., 2013b*; *Cooper, Trinh & Songer, 2009*). Immunizing with both α-toxoid and NetB toxoid provides better protection than when either antigen is used alone (*Fernandes da Costa et al., 2016*). Based on these observations, we designed a novel fusion protein combining toxoids of both toxins. The use of a plant expression system to produce this antigen has many advantages, including high yields, ease of stockpiling, rapid and easy scaling, quick response to emerging pathogens, and lack of any animal components in the final vaccine product.

The production of a PlcC-NetB fusion was readily accomplished in *N. benthamiana* using a geminiviral replicon transient expression system. A previously described *Salmonella* expression system was used to attempt production a fusion protein between PlcC and GST-NetB. However, production levels were low and the protein was rapidly degraded within the *Salmonella* cells (*Jiang et al., 2015*). In contrast, the plant expression system was shown to produce the PlcC-NetB fusion protein at a high capacity without degradation. High levels of protein production were due to our use of the geminiviral replicon transient expression system which has been recently optimized and has undergone improvement since the production of the PlcC-NetB fusion protein (*Diamos, Rosenthal & Mason, 2016*; *Diamos & Mason, 2018*; *Rosenthal, Diamos & Mason, 2018*). Acid precipitation (a process by which the PI of proteins, and their repulsive electromagnetic forces, equal zero and results in protein aggregation) proved to be an effective clarification step, over other purification methods, as both the PlcC-NetB and PlcC-NetB-NG fusion proteins proved to be acid stable (*Wilken & Nikolov, 2012*). This allowed for easy purification using metal affinity chromatography because many of the plant protein contaminants were removed from solution while the fusion proteins remained soluble. Metal affinity chromatography resulted in substantial purification although insubstantial contaminants could be seen upon concentration of elutions (Fig. 3A). However, contaminants were deemed not to be an issue for vaccination as they were in such low quantities.

Since a plant expression system was used, native plant glycosylation of the fusion protein PlcC-NetB occurred. An illustration of this can be seen in both Figs. 2 and 3 as there are progressively higher bands above the primary fusion protein band at 49 kDa only in the ER targeted variant. Glycosylation was a concern as an immune response could potentially be directed toward the glycans rather than the fusion protein itself (*Maverakis et al., 2015*; *Schahs et al., 2007*). This is the reason the PlcC-NetB-NG fusion was pursued. PlcC-NetB-NG proved to be troublesome as it precipitated after dialysis, a phenomenon that has been observed for rubisco, the most abundant plant protein contaminant. Insolubility of PlcC-NetB-NG may also be caused by stripping of metal ions from both PlcC and NetB that are normally present. According to the crystal structure of PlcC it likely binds calcium ions, however, due to difficulty in forming a crystal

structure calcium was substituted for cadmium (*Naylor et al., 1998*). The crystal structure of native NetB toxin has been shown to bind magnesium (*Yan et al., 2013*). These metal ions could be removed during the final dialysis step, which might affect protein structure and solubility. This may indicate that the glycoforms play a role in protein stability and solubility as PlcC-NetB did not have solubility issues under the same buffer, concentration, and purification conditions. Results indicate that, even with plant native glycosylation, the PlcC-NetB fusion promoted an anti-toxoid response. We speculate that the glycoforms could have possibly had an adjuvant effect as antibody titers for the fusion protein were greater than *E. coli* produced PlcC and NetB delivered simultaneously (*Bosch & Schots, 2010*). This may also be due to the fusion of both proteins as well. In any case, production of the PlcC-NetB fusion protein was found to be efficient on a small scale which, due to the nature of plant expression systems, is readily scalable. Moreover, the fusion protein vaccine was found to be effective when injected into broiler birds as is discussed below.

In Experiments 1 and 2, 3 weekly injections with the fusion protein generated strong serum IgY responses against both toxin components (Fig. 4). Vaccinated birds had geometric mean titers of around 2,000 against recombinant NetB and PlcC proteins purified from *E. coli* (Fig. 4). Titers against the PlcC-NetB fusion protein produced in plants were around 8,000 in immunized birds. The difference in titers between single proteins and the fusion protein is speculated to be due to enhanced immune response to the fused protein, since PlcC and NetB are in close proximity during infection, or responses against the carbohydrate moieties on the glycosylated protein leading to an enhanced immune response.

In Experiment 3 birds were immunized at 7 and 12 days of age which lead to a reduced serum response. The chicken adaptive immune system does not fully mature until approximately 2 weeks post-hatch, although humoral responses can be detected after immunization with a protein antigen at 12 days of age (*Mast & Goddeeris, 1999*). In this regard, an adjuvanted NetB toxoid was shown to be ineffective when injected into day-of-hatch chicks (*Mot et al., 2013*). Thus, it is possible that immunizing the birds at 7 days of age did not serve as a priming dose, such that the immunization at 12 days of age was effectively the prime. Previous work showed that immunization at 12 days of age, but not 7 days of age, can be boosted by injection at 29 days of age (*Mast & Goddeeris, 1999*). Although we did not perform a vaccine boost, we infer that the birds were adequately primed since, despite the low serum antibody responses, they were protected against challenge (Fig. 5C).

## CONCLUSIONS

We have demonstrated that this novel toxoid antigen produced in plants is immunogenic and protective in chickens. However, we acknowledge that a more practical immunization approach is needed in order to have a positive impact on the broiler industry. We envision several options. Immunization of parent stock is likely to induce maternal antibodies to protect chicks during the first few weeks of life. This approach, using an α-toxoid antigen, provided partial protection against NE during the first week of

life (*Crouch et al., 2010*). Alternatively, protein antigens can be successfully delivered via in ovo injection with an adequate adjuvant. Delivery of the fusion protein via cornmeal could also be a cheap and effective mass delivery method for vaccine produced in corn seeds. The effectiveness of these approaches will be assessed in future studies.

## ACKNOWLEDGEMENTS

The authors wish to thank Luis Armando Vazquez, Donovan Leigh, Penelope Roach, Melody Yeh, Jack Crawford, Matt Contursi, Mary Pardhe, Sara Aman, and Artemio Chaves for their technical assistance. We thank Dustin McAndrew, Randall Dalbey, Jacquelyn Kilbourne, Larisa Gilley and the rest of the ASU animal care staff for expert care of our research animals and procedural assistance.

### Funding

This project was supported by Agriculture and Food Research Initiative Competitive Grant no. 2016-67016-24947 from the USDA National Institute of Food and Agriculture and startup funds from ASU to Ken Roland, and by support from the Biodesign Institute at ASU for Hugh Mason. The funders had no role in study design, data collection and analysis, decision to publish, or preparation of the manuscript.

### Grant Disclosures

The following grant information was disclosed by the authors:
Agriculture and Food Research Initiative Competitive: 2016-67016-24947.
USDA National Institute of Food and Agriculture and startup funds from ASU.
Biodesign Institute at ASU.

### Competing Interests

K. Roland, A. Diamos, and H.S. Mason are inventors on patents regarding the necrotic enteritis vaccine described herein:

US 62/528,696 JULY 5, 2017.
PCT PCT/US2018/040632 JULY 2, 2018.
Otherwise, the authors declare that they have no competing interests.

### Author Contributions

- Joseph G.L. Hunter conceived and designed the experiments, performed the experiments, analyzed the data, prepared figures and/or tables, authored or reviewed drafts of the paper.
- Shyra Wilde performed the experiments, analyzed the data, prepared figures and/or tables.
- Amanda M. Tafoya performed the experiments.
- Jamie Horsman performed the experiments.
- Miranda Yousif performed the experiments.

- Andrew G. Diamos conceived and designed the experiments, performed the experiments, analyzed the data, contributed reagents/materials/analysis tools, authored or reviewed drafts of the paper.
- Kenneth L. Roland conceived and designed the experiments, analyzed the data, contributed reagents/materials/analysis tools, prepared figures and/or tables, authored or reviewed drafts of the paper, approved the final draft.
- Hugh S. Mason conceived and designed the experiments, analyzed the data, contributed reagents/materials/analysis tools, prepared figures and/or tables, authored or reviewed drafts of the paper, approved the final draft.

### Animal Ethics

The following information was supplied relating to ethical approvals (i.e., approving body and any reference numbers):

Arizona State University Institutional Animal Care and Use Committee approved this research (16-1480R).

### Patent Disclosures

The following patent dependencies were disclosed by the authors:

PCT/US18/40632, VACCINE FOR PREVENTION OF NECROTIC ENTERITIS IN POULTRY

US 13/060,414, A DNA replicon system for high-level rapid production of vaccines and monoclonal antibody therapeutics in plants

### DNA Deposition

The following information was supplied regarding the deposition of DNA sequences:

The fusion protein nucleotide sequences are in GenBank accession numbers MH883808 and MH883809.

### Data Availability

Original Western blot images for Figs. 2 and 3 are available in the Supplemental Materials.

### Supplemental Information

Supplemental information for this article can be found online at http://dx.doi.org/10.7717/peerj.6600#supplemental-information.

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
