# Peer review of "Evaluation of a toxoid fusion protein vaccine produced in plants to protect poultry against necrotic enteritis"

_PeerJ, doi:10.7717/peerj.6600_

## Round 0.1 · original submission · Minor Revisions

I strongly encourage you to make the minor modifications suggested by reviewers 2 and 3, to increase the clarity of your manuscript.

·

Basic reporting

Evaluation of a toxoid fusion protein vaccine produced in plants to protect poultry against necrotic enteritis. The authors developed a fusion protein combining a non-toxic carboxyl-terminal domain of atoxin (PlcC) and an attenuated, mutant form of NetB (NetB-W262A) for use as a vaccine antigen to immunize poultry against NE, using a DNA sequence that was codon-optimized for Nicotiana in broilers. Immunized birds produced a strong serum IgY response against both the plant produced PlcCNetB protein and against bacterially produced His-PlcC and His-NetB and were significantly protected against a subsequent in-feed challenge with virulent C. perfringenswhen treated with the suggesing that a plant-produced PlcC-NetB toxoid is a promising vaccine. The article is well written and clear and has sufficient field background/context. It has professional structure, figs, and tables.

Experimental design

The research queestion is well defined and the experimental design well established. Authors perfomred a rigorous research and provide in their material and methods section, sufficient detail so other researchers could replicate their work.

Validity of the findings

The study is novel and the reuslt resented are robust, statistically sound, and controlled. Their conclusion are well stated, linked to original research question limited to supporting results.

Additional comments

Study well conducted and written.

·

Basic reporting

the authors have designed a fusion gene expressed in plant leaves for the production of netB an alpha toxoid, which was shown to be immunogenic and partially protective when injected into broilers. Literature references in the introduction should be used with care. Refering to the mouse model of gas gangrene is of no use in the context of necrotic enteritis in broilers.Not all references have shown documented protection against necrotic enteritis lesions, and if they did, protection was always partial.
Parts of the materials and methods are mentioned in the results section.
Line 352: Fig. 5 should read Fig. 6.
the authors mention high yield as an advantage of their expression system, however, the yield is not reported in the results section.
Line 388: NG mentioned for the first time. What is this?
Line 430: Please read paper 29 carefully: effectiveness is questionable.
Relevance of results: the progress compared to what is reported in literature with both antigens separately in other expression systems is limited, whit actually no difference in protection.

Experimental design

Research question is not clearly defined. This is the application of different currently available tools to improve immunogenicity of vaccine antigens against necrotic enteritis in broilers.
The methods are described with sufficient detail.

Validity of the findings

Data and statistics are sufficiently sound.
Novelty lies only in the application of the plant expression system to C. perfringens antigens.
conclusions are well stated.

·

Basic reporting

Clear and unambiguous, professional English. Well written and well designed experiments.

Experimental design

Well designed experiments well within the scope and aims of the journal. The only issue I found was that the number of birds/treatment was not include din the text of the M&M. Otherwise, the experimental detail was more than sufficient.

Validity of the findings

All findings were documented and repeated. The results were solid and believable.

---

## Round 0.2 · accepted · Accept

I appreciate your taking into account all comments raised by reviewers.

# ·

Basic reporting

the authors responded to the comments correctly.
I have no further comments.

Experimental design

O.K.

Validity of the findings

O.K.

Additional comments

no further comments